# Modelling transmission thresholds and hypoendemic stability for onchocerciasis elimination

Jacob N. Stapley[1]*, Jonathan I.D. Hamley[1,2,3], Maria-Gloria Basáñez[1‡],
Martin Walker[1,4‡]

1 MRC Centre for Global Infectious Disease Analysis and London Centre for Neglected Tropical Disease Research, Department of Infectious Disease Epidemiology, School of Public Health, Imperial College London, London, United Kingdom, 2 Department of Visceral Surgery and Medicine, Inselspital, Bern University Hospital, University of Bern, Bern, Switzerland, 3 Multidisciplinary Center for Infectious Diseases, University of Bern, Bern, Switzerland, 4 Department of Pathobiology and Population Sciences, Royal Veterinary College, Hatfield, United Kingdom

‡Joint senior authors.
* j.stapley20@imperial.ac.uk

## Abstract

The World Health Organization (WHO) has proposed elimination of onchocerciasis transmission (EOT) in a third of endemic countries by 2030. This requires country-wide verification of EOT. Prior to the shift from morbidity control to EOT, interventions in Africa were mostly targeted at moderate- to high-transmission settings, where morbidity was most severe. Consequently, there remain numerous low transmission (hypoendemic) settings which have hitherto not received mass drug administration (MDA) with ivermectin. The WHO has prioritised the delineation of hypoendemic settings to ascertain treatment needs. However, the stability of transmission at such low levels remains poorly understood. We use the stochastic EPIONCHO-IBM transmission model to characterise the stability of transmission dynamics in hypoendemic settings and identify a range of threshold biting rates (TBRs, the annual vector biting rates below which transmission cannot be sustained). We show how TBRs are dependent on population size, inter-individual exposure heterogeneity and simulation time. In contrast with deterministic expectations, there is no fixed TBR; instead, transmission can persist between 70 and 300 bites/person/year. Using survivorship models on data generated from model simulations, we find that multiple vector biting rates can sustain hypoendemic prevalence for several decades. These findings challenge the assumption that hypoendemic foci would naturally fade out following treatment in nearby higher-endemicity regions. Our modelling suggests that, to achieve EOT, treatment should be extended to all areas where endogenous infection is identified, emphasising the need for improved diagnostic tools suitable for detecting low-prevalence infection and for strategies that allow safe treatment of communities where MDA would not be suitable.

**Data availability statement:** This was an in-silico study which did not utilise raw data for the modelling presented here. Previously published epidemiological data were used as means of comparison to the model outputs. These are contained within the manuscript, its references and its supporting information files. The model code can be found at: https://github.com/mrc-ide/EPIONCHO.IBM at https://doi.org/10.1038/s41467-024-50582-9.

**Funding:** J.N.S. received funding from the UK Medical Research Council 1+3 MRes+PhD Studentship in Infectious Disease Epidemiology (MR/S502388/1). M.G.B. acknowledges funding from the MRC Centre for Global Infectious Disease Analysis (MR/X020258/1), funded by the UK Medical Research Council. This UK-funded award is carried out in the frame of the Global Health EDCTP3 Joint Undertaking. M.G.B. and M.W. also acknowledge funding by the Gates Foundation through the NTD Modelling Consortium (INV-030046). The funders had no role in study design, data collection and analysis, decision to publish, or preparation of the manuscript.

**Competing interests:** The authors have declared that no competing interests exist.

## Author summary

Onchocerciasis (river blindness) is a filarial infection transmitted among humans via the bites of blackfly vectors and targeted for elimination of transmission by the World Health Organization. While control efforts in Africa have hitherto focused on areas with moderate–high transmission, many of those with low transmission (hypoendemic areas), have not yet been incorporated into mass ivermectin treatment programmes. Using a detailed computational model, we investigated whether transmission in these low-endemicity settings will naturally disappear or might persist for a considerable time. We show that even in areas with low blackfly biting density, transmission can continue for decades at very low infection levels. Our results suggest that to achieve elimination of onchocerciasis transmission, appropriate treatment strategies (mass treatment or more targeted approaches) must be expanded to all areas where locally-derived infection is identified. This work also emphasises the importance of developing improved diagnostic tools capable of detecting infections in low-transmission communities. Our findings demonstrate how computational modelling can help guide public health decision-making for disease elimination.

## Introduction

Onchocerciasis is a filarial infection caused by *Onchocerca volvulus* and transmitted among humans by *Simulium* blackfly vectors. The Global Burden of Disease Study estimated that in 2021, 20 (95% Uncertainty Interval (UI) = 18–22) million people were infected and that the disease was responsible for 1.26 (95% UI = 0.75–1.90) million disability-adjusted life years (DALYs) [1]. The great majority of cases (>99%) are in sub-Saharan Africa (SSA). The microfilarial (mf) progeny of adult worms are the stages transmitted from humans to vectors and mostly responsible for the clinical manifestations of onchocerciasis, including dermatological conditions, visual impairment, and blindness [2]. Chronic, untreated infection with *O. volvulus* is also associated with epilepsy and significant excess mortality following a dose-dependent relationship, particularly in children [3–5].

Long-term interventions, principally based on mass drug administration (MDA) of ivermectin, have eliminated the disease from four countries in Latin America and reduced its transmission across SSA [6]. In its 2021–2030 Roadmap on Neglected Tropical Diseases, the World Health Organization (WHO) proposed that elimination of onchocerciasis transmission (EOT) be verified in approximately a third of endemic countries by 2030 [7]. This requires that country-wide EOT be achieved in all foci. Hypoendemic foci (those with mf prevalence less than 35%) had not been prioritised for ivermectin MDA as they were deemed not to contribute to (particularly ocular) morbidity [8]. Rapid epidemiological mapping of onchocerciasis (REMO), used prevalence of palpable onchocercal nodules in samples of adult men to identify high-risk (meso- and hyperendemic) areas (those with mf

prevalence ≥35% or nodule prevalence ≥20%) for ivermectin MDA [8,9]. With the policy shift from morbidity control to EOT initiated in 2012 [10], greater emphasis has been placed on understanding and delineating low-transmission, hypoendemic areas.

To this end, the WHO recommends that onchocerciasis elimination mapping (OEM) be undertaken, firstly to identify potentially suitable transmission areas and presence of vectors, and second to conduct serological surveys in resident adults aged ≥20 years. Cut-off values for initiation of MDA have been proposed according to the diagnostic tool used to detect IgG4 antibody seropositivity against the *O. volvulus* Ov16 antigen, with cut-off values representing the number of seropositive individuals found in a village (from 1 to 4 depending on diagnostic test, sampling protocol and OEM step, having sampled a minimum of 100 individuals per village) [11].

Notwithstanding, the stability of transmission in hypoendemic foci is a longstanding and unanswered question in onchocerciasis epidemiology. The decision to focus MDA on meso- and hyperendemic foci by the African Programme for Onchocerciasis Control (APOC, 1995–2015) was in part because APOC was initially conceived as a disease control (not an EOT) programme [12] but also based on the notion that hypoendemic foci may not be endemically stable without regular importation of infections from nearby higher-endemicity areas [13]. That is, in hypoendemic settings the annual biting rate (ABR, no. bites/person/year) of blackfly vectors may be insufficient to sustain endogenous (locally-derived) transmission. Hence, as reductions in transmission are achieved in meso- and hyperendemic settings, the expectation would be that infection likely 'fades out' in proximate hypoendemic settings.

Mathematical models have been used to estimate threshold biting rates (TBRs; ABRs below which transmission cannot be sustained). These models have mainly been deterministic, such that a TBR is uniquely defined by a fixed set of model parameters and serves as a threshold quantity below which transmission is unsustainable (the basic reproduction ratio, $R_0$, is < 1) [14–16]. By contrast, individual-based stochastic models encapsulate probabilistic (stochastic) events that become increasingly important for small human and/or parasite population sizes [17–19]. For filarial infections, such models predict that for a given TBR there will be a distribution of outcomes corresponding to endemicity or local extinction [17,20] (see Fig A in S1 Text for an illustration).

We have previously used the stochastic individual-based onchocerciasis transmission model, EPIONCHO-IBM, to investigate ABRs required to generate a stable equilibrium mf prevalence in 100% of model simulations [17,21]. However, our focus has mostly been on the ABR–mf prevalence relationship in meso- to hyperendemic settings [17]. Here, we use EPIONCHO-IBM to characterise the transmission dynamics of onchocerciasis in hypoendemic settings near TBRs. We evaluate the stability of mf prevalence using the rate of fade-out of model simulations to define the probability and likely duration of persistence over epidemiologically-relevant time horizons. We also consider the effect of different host population sizes and inter-individual heterogeneity in exposure to blackfly bites. We discuss our findings in the context of decision-making on whether to initiate MDA in newly identified hypoendemic settings.

## Methods

### EPIONCHO-IBM

We used the EPIONCHO-IBM individual-based, stochastic onchocerciasis transmission model [17]. The model tracks the number of adult *O. volvulus* worms in each individual host in the human population; their skin mf load, and the mean no. of infective, L3 larvae per (*S. damnosum* sensu lato (s.l.)) blackfly. Density-dependent processes regulate parasite population abundance in both humans and vectors [14,15]. These processes upregulate or downregulate parasite transmission when their probabilities (or rates) increase or decrease with parasite density, respectively. Examples of the former in EPIONCHO-IBM include the proportion of female worms mated in the presence of male worms (the mating probability), and of the latter, the proportion of L3 larvae that successfully develop into adult worms within the human host [17]. Individuals within the population are differentially exposed to blackfly bites depending on their age, sex, and an individual-specific exposure factor drawn from a gamma distribution, with (shape and rate) parameter $k_E$. The lower the magnitude of

$k_E$, the stronger the degree of exposure heterogeneity among individuals (Text A and Table A in S1 Text). A full description of the model can be found in [17].

### Initial conditions

We used a burn-in period of 80 years to generate stable initial conditions, testing ABRs between 250 and 40,000 bites/person/year, ensuring negligible probabilities of stochastic fade-out. Once this equilibrium was established, we set time to $t=0$ days and dropped the ABRs to the TBR range under consideration (informed by [17]), running simulations from $t=0$ days for 500 years repeated 300 times. A 500-year simulation period was chosen to be an arbitrary long timeframe to evaluate comprehensively the dynamics of persistence and stochastic fade-out over time, whilst remaining within computationally accessible limits. Similarly, the 300 repeats were chosen to ensure that persistence could be detected even at an arbitrarily low probability (1/300) and provide a near-continuous scale on the relationship between persistence probability and ABR. The magnitude of the ABRs used to generate the initial equilibrium conditions—prior to the drop in ABR—did not alter the proportion of simulations persisting, nor their mean mf prevalence after the 500-year simulation period (Fig B in S1 Text).

### Threshold biting rates

We defined a TBR range as the ABRs for which a proportion (>0% and <100%) of 300 repeat model runs remained above 0% mf prevalence (measured across the total population) after the 500-year simulation period. Model runs generated by an ABR above the TBR range would 'always' persist (strictly, with probability >299/300) whereas below the TBR range, they would 'never' persist (strictly, with probability <1/300). When the ABR is within the TBR range, model runs persist with a non-degenerate probability distribution (i.e., with probability values between 'always persisting' and 'never persisting'). We investigated how the proportion of runs persisting over the simulation period evolved over time. This was done by varying the human population size $N$ from 50 to 1000 (values typical of endemic villages across SSA [13,22]) and the magnitudes of inter-individual exposure heterogeneity from $k_E=0.2$ to $k_E=0.4$, using corresponding density dependence parameters, with values informed by previous model fitting to paired parasitological and entomological data [17] (Table A in S1 Text).

### Rates of fade-out

Survivorship (exponential, Gompertz, and Weibull) models were fitted to the proportion of model runs persisting over the simulation period, calculated at each 5-year interval. A Weibull model (Eqn. 1), provided the best fit across the TBR range (based on minimising the sum of squared residuals),

$$P(t) = exp(-at^b) \tag{1}$$

where $P(t)$ is the proportion of runs persisting at time $t$ (in days), and $a$ and $b$ represent the scale and shape parameters, respectively. The Weibull model was fitted from the final timepoint at which 100% of model runs persisted (Fig C in S1 Text).

The estimated parameter values arising from fitting the Weibull model were used to calculate the rate of fade-out (equivalent to the hazard rate, i.e., the instantaneous risk of fade-out) using Eqn. (2),

$$h(t) = \left(\frac{b}{a}\right)\left(\frac{t}{a}\right)^{(b-1)} \tag{2}$$

All analyses and data visualisations were undertaken using R, version 4.3.2 [23]. The model code is available at: https://github.com/mrc-ide/EPIONCHO.IBM, as referred to in [24].

We adhered to the five principles of the Neglected Tropical Diseases (NTD) Modelling Consortium regarding Policy-Relevant Items for Reporting Models in Epidemiology of NTDs (PRIME-NTD), for good practice in NTD modelling [25] (Text B and Table B in S1 Text).

## Results

### Stability of hypoendemic settings

Commencing from initial conditions with 100% of model simulations persisting (using an ABR of 1,000 bites/person/year), followed by decreasing ABR to identify TBR ranges, runs become extinct over time. The proportion of persisting runs is dependent on elapsed simulation time. For example, for $N=400$, $k_E=0.3$ and ABR $=180$, approximately 90% of runs persist after 110 years, approximately 30% persist after 220 years and less than 10% persist after 500 years (Fig 1A).

### Threshold biting rates

At the end of the 500-year simulation period (Fig 1B) the proportion of persistent runs yielding mf prevalence >0%—defining our TBR range—depends on individual heterogeneity in exposure, $k_E$, and the human population size, $N$. For example, for $k_E=0.3$ and $N=400$, the TBR ranges between 180 and 240 bites/person/year (Fig 1B; $N=400$, $k_E=0.3$). A decrease in $N$ from 400 to 50 individuals increases the minimum TBR from 180 to 200 bites/person/year alongside decreasing the proportion of persisting runs at any given ABR (Fig 1B; $N=50$, $k_E=0.3$). By contrast, an increase in $N$ from 400 to 1000 individuals increases stability, resulting in a higher proportion of persisting runs for a given ABR (Fig 1B; $N=1000$,

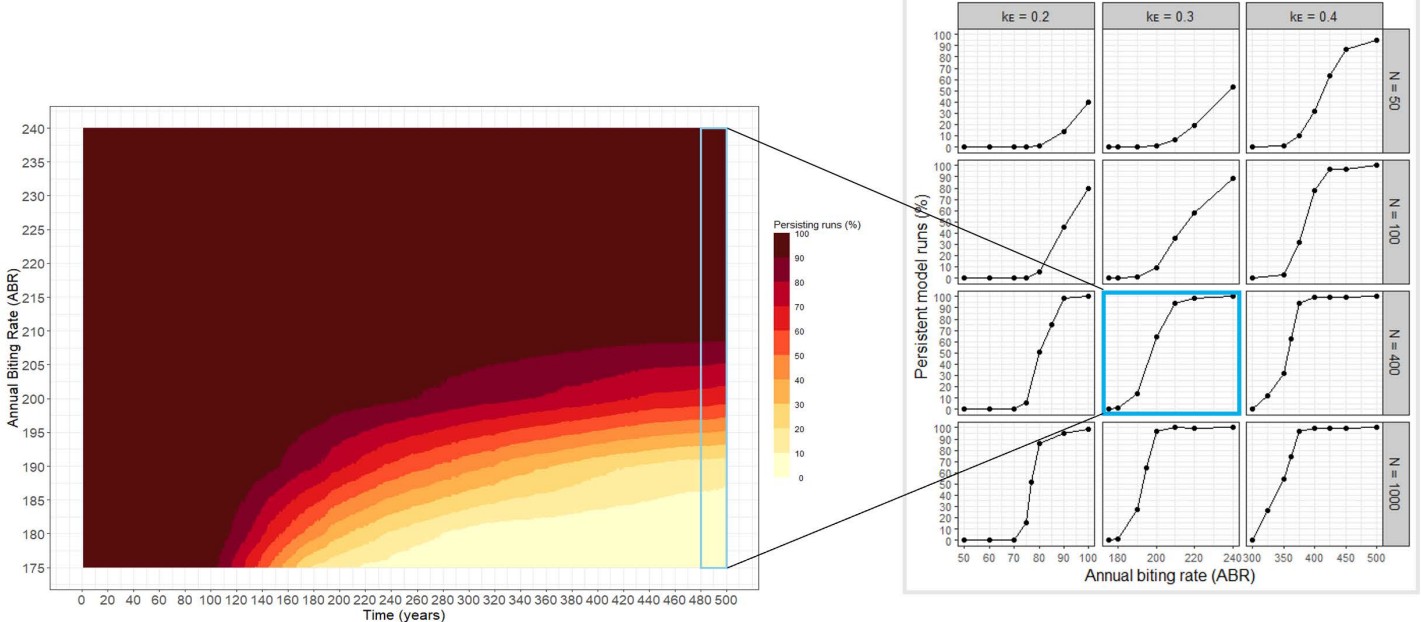

**Fig 1. Stability of hypoendemicity and threshold biting rates (TBRs).** Three hundred repeat simulations of EPIONCHO-IBM were run for 500 years, determining the proportion of persistent runs with greater than 0% microfilarial (mf) prevalence. **Panel A:** Colours from dark red to light yellow indicate a decreasing proportion of model simulations persisting over time for different annual biting rate values (ABRs). Results are shown for a population of 400 individuals ($N=400$) and inter-individual exposure heterogeneity parameter $k_E=0.3$, with the light blue box highlighting persistent proportions after 500 years of simulation. **Panel B:** Markers are the proportion of the 300 model runs persisting after 500 years for ABRs falling within the TBR range (i.e., >0% and <100% run persistence). The blue-highlighted box shows the proportion of persistent runs at the simulation endpoint as generated by the model parameters used in Panel A. Remaining boxes show different combinations of $N$ and $k_E$ as indicated by the row and column labels, respectively.

$k_E=0.3$). Greater inter-individual exposure heterogeneity ($k_E=0.2$) is associated with decreasing minimum TBR values compared to $k_E=0.3$ for all population sizes. For example, for $N=400$, the minimum TBR of 180 bites/person/year for $k_E=0.3$ decreases, by roughly 60%, to approximately 75 bites/person/year for $k_E=0.2$ (Fig 1B; $N=400$, $k_E=0.3$ vs. $N=400$, $k_E=0.2$). The opposite is true when decreasing inter-individual exposure heterogeneity (e.g., from $k_E=0.3$ to $k_E=0.4$).

## Microfilarial prevalence dynamics and persistence

The dynamics of simulation fade-out is well characterised by a Weibull survivorship model (Fig C in S1 Text), which were used to estimate the time-dependent, instantaneous rate (risk) of fade-out (i.e., the hazard). The rate of fade-out depends on the ABR and increases non-linearly with time, initially increasing rapidly before slowing down (Fig 2) and resulting in a substantial distributional tail, with few simulations persisting for a long time (Fig C in S1 Text). The distributions of mean mf prevalence across the 300 repeat simulations over time, and for different $N$ and $k_E$ are shown in Figs D–F in S1 Text, highlighting how persistent simulations are distributed around a hypoendemic equilibrium.

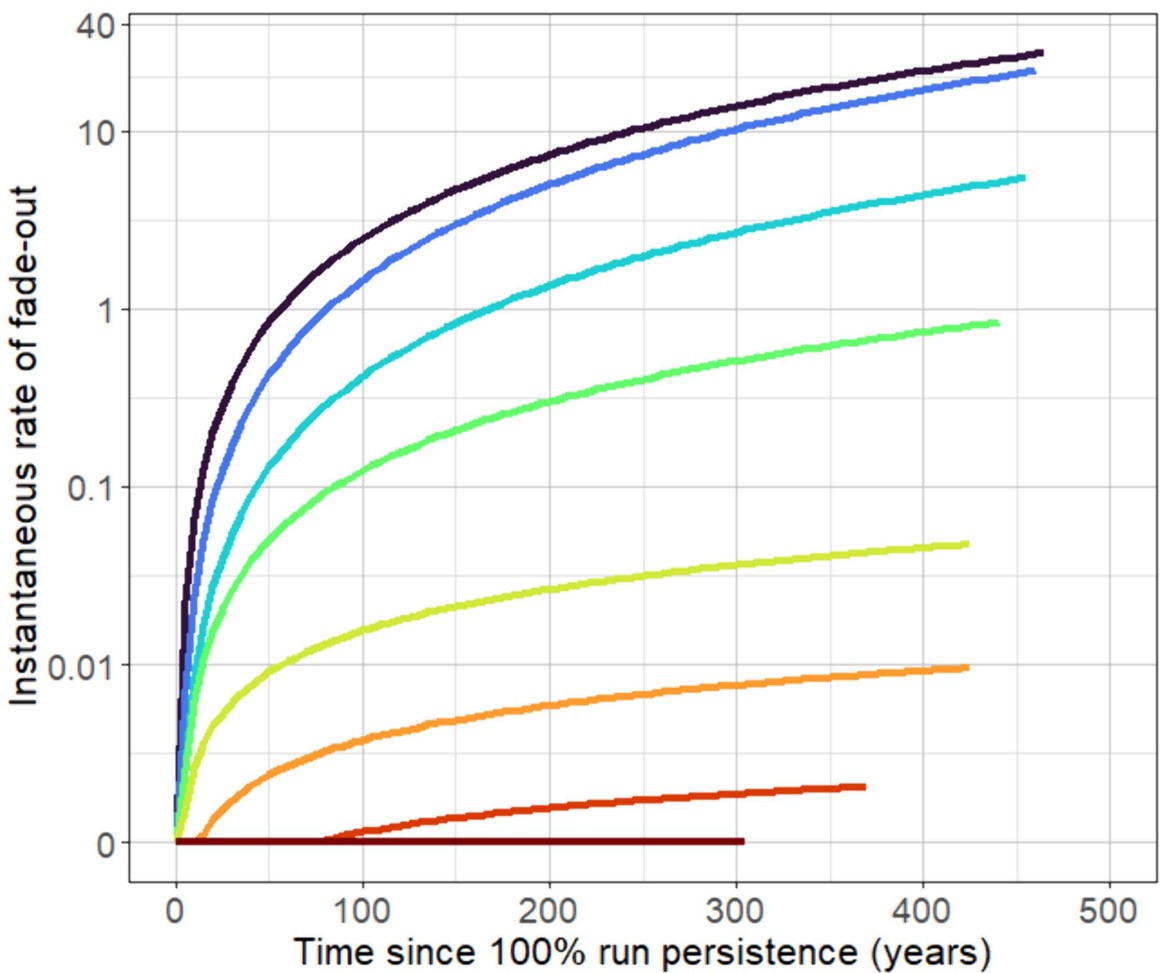

**Fig 2. Rate of fade-out of onchocerciasis transmission within and below threshold biting rate (TBR) range.** The colour of each line corresponds to an annual biting rate (ABR; no. bites/person/year) for a population of $N=400$ and $k_E=0.3$, with the rate of fade-out calculated by fitting a Weibull survivorship function (Eqn. 1, 2) to 300 EPIONCHO-IBM simulations (Fig C in S1 Text) from the final timepoint which gave 100% run persistence (i.e., all simulations > 0% microfilarial prevalence).

For an ABR of 210 ($N=400$, $k_E=0.3$), the rate of fade-out across the simulation period remained approximately zero (Fig 2) with a mean mf prevalence of 16% after 500 years. The mf dynamics from 16% mf prevalence over a period of 150 years are shown in Fig 3 for several ABRs both within and below the TBR range for $N=400$ and $k_E=0.3$. For an ABR of 200, the mf prevalence drops from 16% to 12% over 150 years, indicating a high degree of stability. By contrast, for an ABR of 180—which is at the lower end of the TBR range for $N=400$ and $k_E=0.3$—the mf prevalence drops, more rapidly, to <2% after 150 years. Below an ABR of 180—which is below our identified TBR range—mf prevalence drops more precipitously, halving to 8% in approximately 15–30 years (for ABRs ranging from 100 to 160), albeit remaining persistent over this epidemiologically-relevant time period.

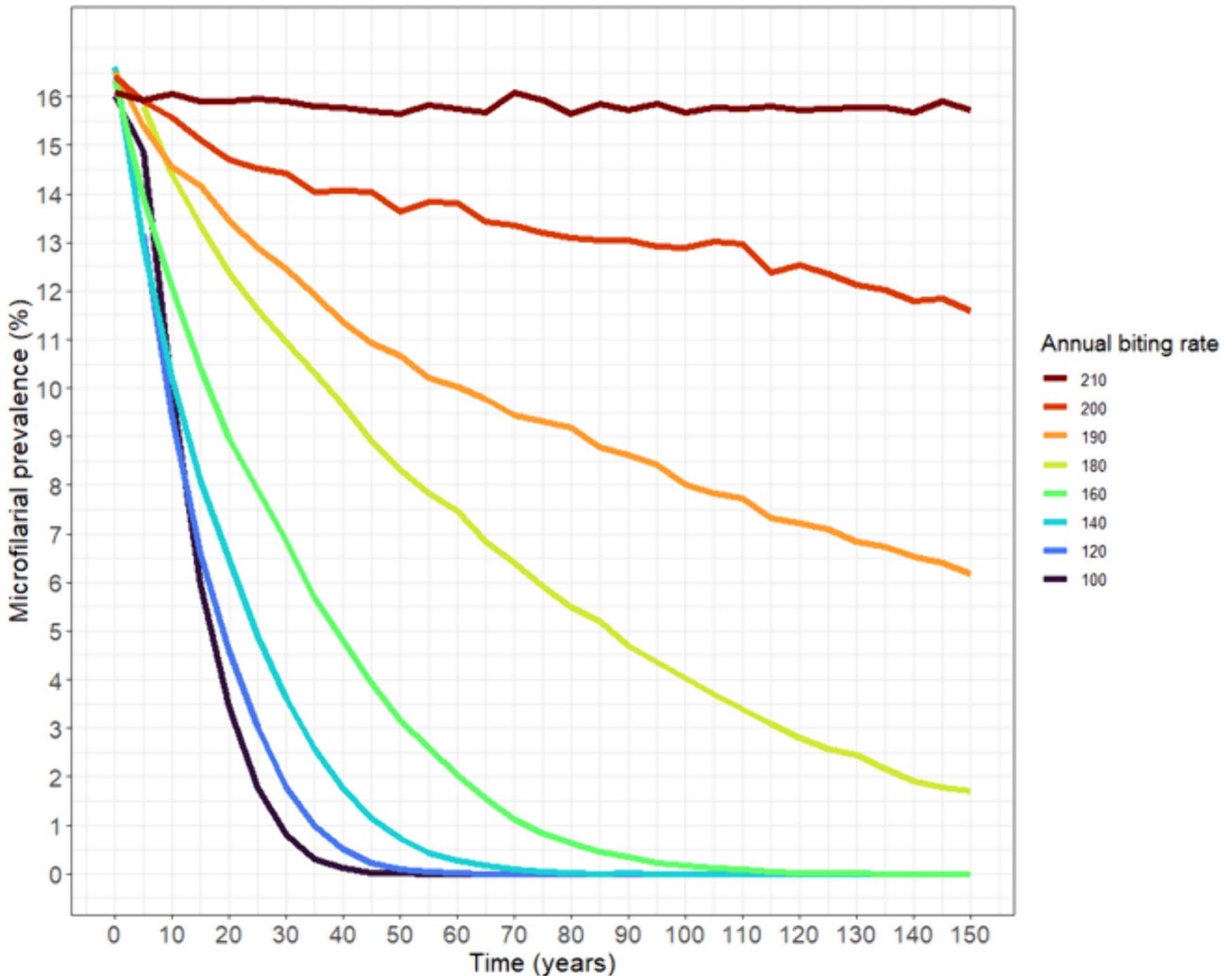

**Fig 3. Mean microfilarial (mf) prevalence dynamics within and below threshold biting rate (TBR) range.** The colour of each line corresponds to an annual biting rate (ABR; no. bites/person/year) for a population of $N=400$ and $k_E=0.3$, with mean mf prevalence calculated from 300 EPIONCHO-IBM simulations. Dynamics are plotted from a 16% mf prevalence at time zero corresponding to the minimum mf prevalence with no fade-out over 500 years (i.e., zero rate of fade-out; Fig. 2). Mf prevalence declines more rapidly with decreasing ABR, but may persist (i.e., mf prevalence >0%) for epidemiologically-relevant periods even at very low ABRs.

## Discussion

We have explored the stability and stochastic dynamics of onchocerciasis in low-transmission hypoendemic settings using the EPIONCHO-IBM individual-based transmission model [17]. We show that the stability of transmission is highly sensitive to the ABR, which is exacerbated by small human population sizes, but stabilised by increasing degrees of inter-individual heterogeneity in exposure. Therefore, the stability of transmission is a function of ABR, human population size, exposure heterogeneity and simulation time, with the chance of spontaneous stochastic fade-out increasing over longer time horizons. Thus, in a classical deterministic sense, there is no simple definition of a TBR. However, by estimating the long-term persistence of transmission after a nominal 500 years, we define ranges of notional TBRs between approximately 70 and 300 bites/person/year, depending on human population size and inter-individual exposure heterogeneity. We have also explored rates of fade-out and declines in mf prevalence, within and below our TBR range, to determine for how long settings with hypoendemic and unstable transmission may persist. These findings have practical implications for decision-making on whether to initiate treatment in low-endemicity settings which should consider the likelihood that newly identified hypoendemic settings will persist for epidemiologically-relevant timeframes without intervention.

Our notional TBR range is approximately concordant with previous deterministic and hybrid modelling approaches, which estimated TBRs for *S. damnosum* s.l. blackflies in West African savannah settings between 300 and 730 bites/person/year [15,16]. These values were dependent on the assumed proportion of human blood-meals taken by the vectors (here assumed to be approximately two in every three) [17], with different vector species having varying degrees of anthropophagy (propensity to feed on humans) [26] as well as vector competence for *O. volvulus*. Higher anthropophagy and vector competence would yield lower TBRs and vice versa [14–16,27,28]. EPIONCHO-IBM incorporates parasite density-dependent population processes in humans and vectors [15,17,27,29,30] which increase stability and enhance resilience to intervention [17,31,32] compared to other transmission models [30,33,34], which explains our low TBRs. Since we also explored the influence of strong exposure heterogeneity, our TBR range is somewhat lower than previous estimates. The assumed operation of these processes also enables modelling of very low mf prevalence, which are stable over epidemiologically-relevant timescales without importation of infection. The long lifespan of adult *O. volvulus* (we assume a maximum lifespan of 20 years with an average of 9–11 years [17,35]) facilitates protracted persistence which will be longer than for other helminths with substantially shorter lifespans (see [19] for an example in shorter lived helminths).

Our findings are consistent with epidemiological observations at focus, national and regional levels in SSA. At focus level, a study in Cameroon, aimed at investigating whether autochthonous transmission occurs in hypoendemic areas, conducted a survey of 10 ivermectin-naïve communities situated along the river Mayo Douka in the Ngong Health District of Cameroon. The villages shared similar ecological and entomological conditions and there was no migration (participants were resident in their communities for at least 10 years prior to the study, and the villages were >20 km away from the nearest meso-⁄hyperendemic zone). The mf prevalence in adults (aged >20 years) ranged from 0% to 12%, and in children (aged 3–10 years) ranged from 0% to 2%. The average ABR over two years was 178 (within our TBR), with 0.016 L3 larvae per fly (indicative of active transmission). The authors concluded that Ngong is a hypoendemic focus with endogenous transmission in isolation from meso- and hyperendemic areas [13].

At national level, a systematic review, covering studies across a 26-year period from 1992-2018 in Gabon, reported nationwide microfilarial prevalence fluctuating between 42% in 1994 and 8% in 2006 in the absence of any control measures [36]. In 2015, of 31 surveyed districts, only four exhibited meso- or hyperendemicity, while the majority (87%) indicated hypoendemicity according to mf prevalence [37]. Neither ivermectin MDA nor vector control against onchocerciasis have ever been implemented in Gabon, suggesting that hypoendemicity can persist over decades.

At regional level, our results are further supported by pre-intervention epidemiological data collected under the Onchocerciasis Control Programme in West Africa (OCP). This dataset comprised baseline mf prevalence from 737 villages (before the initiation of control interventions) [22]. Among these, 172 villages (23%) were classified as hypoendemic, with

mf prevalence ranging between ≥10% and <35%, indicating that hypoendemicity was not a rare phenomenon in West Africa, attesting to the occurrence of stable hypoendemic transmission. Notably, O'Hanlon *et al.* (2016) [22] categorised <10% microfilarial prevalence as 'non-endemic/sporadic endemicity', a stratification which aligns well with our modelling projections. In Fig 3 of our study, we show that biting rates generating <10% prevalence fall below the threshold biting rate (TBR) range, making them susceptible to spontaneous fade-out, albeit this could take place over many decades.

When calibrating EPIONCHO-IBM with the Gabon mf data for hypoendemic villages [37] (mean observed mf prevalence = 5.5%; mean modelled mf prevalence = 5%), the best-fit parameters were ABR = 73 and $k_E$ = 0.2 [38], consistent with our results. This is in contrast with other models which are unable to generate low mf prevalence endemic stability at these low ABRs, in closed populations without a rate of infection importation from external sources [34,39] but see [40].

The protracted dynamics of unstable transmission raise important questions for interventions targeting EOT. We see that (when $N$ = 400 and $k_E$ = 0.3) an ABR of 210 generates 16% mf prevalence, the lowest prevalence which does not fade-out over the simulation period. Below 16%, fade-out occurs over time. This aligns with the endemicity stratifications used in O'Hanlon *et al.* [22], where <10% mf prevalence was deemed 'non-endemic/sporadic endemicity.' However, even below 10% mf prevalence (and below our estimated TBR range), the rate of fade-out of infection can be slow. Moreover, multiple ABRs can yield the same mean mf prevalence at a single point in time (cross sectional snapshot), albeit with very different future trajectories. For example, our simulations show that a 9% mf prevalence can result from 6 different ABRs (Fig 3). However, for an ABR of 190, a 9% mf prevalence will persist (albeit declining) for, on average, at least another 100 years. By contrast, a 9% mf prevalence generated from an ABR of 100 will have faded out (gone extinct) within 20 years with very high probability. This makes it challenging to predict the stability of hypoendemic transmission and the likelihood of spontaneous elimination based on cross-sectional surveys of mf prevalence (or any other infection metrics) alone.

Measuring ABR in the field remains challenging, with the WHO Onchocerciasis Technical Advisory Subgroup (OTS) calling for more efficient blackfly capture methods and sampling protocols, as well as for improved molecular tools to detect infection and infectivity in blackfly population samples [41,42]. Besides, because skin-snip microscopy is an invasive and moderately painful diagnostic procedure, with increased reluctance of endemic populations to participate in parasitological surveys [43], and poor sensitivity in low-transmission settings [44], decisions on whether to initiate MDA in newly identified hypoendemic settings are currently based on determining anti-Ov16 seropositivity in adults aged ≥20 years [11]. Integrating anti-Ov16 seroprevalence as a model output of EPIONCHO-IBM is underway [38]. This will be an important next step in the use of modelling for informing decision-making on where MDA is required. However, we anticipate that, like with mf prevalence, single cross-sectional surveys of anti-Ov16 seroprevalence without concurrent estimates of ABRs will be of limited use for making robust predictions on the likely stability of transmission in low-transmission foci.

Our simulations show how, in addition to ABR, human population size and inter-individual heterogeneity in exposure are key parameters influencing stability in low-transmission settings. Increasing population size and increasing the strength of inter-individual exposure heterogeneity increase stability and vice versa for decreasing population size and decreasing inter-individual exposure heterogeneity [19]. Small population sizes are well known to exacerbate stochastic effects—increasing the chance of spontaneous fade-out. Hence, at least qualitatively, larger communities even with very low levels of infection will be more conducive to stable transmission than small communities and, therefore, of higher priority for intervention. Some of these communities may have already received MDA in lymphatic filariasis-onchocerciasis co-endemic areas [11]. However, the issue of starting MDA in hitherto untreated hypoendemic onchocerciasis communities raises important issues when these communities are also co-endemic with loiasis (another filarial infection caused by *Loa loa* and transmitted by *Chrysops* tabanid flies). This is because there is an increased risk of individuals with high mf loads of *L. loa* (>20,000 mf/mL of blood) experiencing severe adverse events if treated with ivermectin [12,45]. Foci with moderate to high levels of loiasis endemicity would not be suitable for MDA. Here, Test and Not Treat (TaNT) strategies using LoaScope technology will be required [45]. Ivermectin MDA may be appropriate in foci with low loiasis endemicity.

In both cases, pharmacovigilance/monitoring will be required [46]. To address these issues, the WHO OTS has suggested incorporating *L. loa* mapping into OEM [47]. Other factors, such as civil unrest or active conflict, may also hinder initiation of MDA, irrespective of local epidemiology.

Our results also highlight the vulnerability of infection-free populations to the importation of infection from larger proximate hypoendemic foci with ongoing low-level transmission [39,48]. Inter-individual variability in exposure to blackflies, albeit of key importance to enhancing stability of onchocerciasis transmission [40], is notoriously difficult to quantify in practice. However, progress has been made in developing antibody assays against blackfly salivary antigens for this purpose [49,50].

**Limitations.** In stochastic frameworks such as the one explored here, fixed TBR values, below which transmission cannot persist do not formally exist. If closed population simulations (i.e., with no importation of infection) were run over an infinite time period, they would eventually all fade out, irrespective of ABR, population size, inter-individual variability in exposure or other parameters. In practice, however, this probability becomes infinitesimally small for larger ABRs. Hence, we took a pragmatic approach to defining TBR ranges, quantifying the probability of persistence over an arbitrarily long 500-year time horizon. Simulations with >0% and <100% chance of persistence during this period were defined as within the TBR range, but we acknowledge that, for instance, using a longer time horizon would increase the lower bound of this range. Equally, a shorter time horizon would indicate lower ABRs being consistent with a non-zero probability of sustaining transmission.

Another important limitation of the current work is that we assume that the annual biting rate (ABR) of blackfly vectors remains constant over time. While this is reasonable because we are not considering vector control interventions—which would certainly reduce ABR—it will not capture secular changes in ABR, related to climate change or other anthropogenic effects (e.g., deforestation, pollution of blackfly breeding sites). Furthermore, it has been hypothesised that there is a likely relationship between ABR and $k_E$, by which $k_E$ would be expected to increase (decreasing exposure heterogeneity) for increasing ABR values. Indeed, previous work [17] has shown that at high ABRs—consistent with hyper- and holendemicity—EPIONCHO-IBM captures better the ABR-microfilarial prevalence relationship using higher values of $k_E$ ($k_E = 0.4$), and vice versa ($k_E = 0.2$ capturing better lower prevalence at low ABRs). However, modelling work fitting EPIONCHO-IBM to OCP village data has only found a modest relationship between these two parameters (Dr Raiha Browning, pers. comm.).

Other factors not explored in this paper will also influence the TBR. The model does not include excess mortality associated with infection in the human population [4,5], which could, in principle, decrease stability and persistence by reducing the lifespan of individuals with high mf loads. However, we consider this effect to be of limited significance in the low-transmission settings considered. More pertinent, is the potential impact of imported infections via immigration which could enhance stability [51]. Here we have assumed a closed population, with stability mediated by (down-regulating) parasite density-dependent processes which enhance the efficiency of transmission at low endemicity. Very little is known on patterns of human and/or blackfly movement among rural endemic communities [39,52] and so it is currently impossible to disentangle the contributions of importation and density dependence on endemic stability. This topic warrants additional research, not only in terms of understanding the causes of low-level endemic persistence, but also critically, the risk of re-introduction of infection where elimination has been reached [39,40,48].

**Conclusions and recommendations.** Our pragmatic choices for defining TBRs reflect the programmatic decisions that public health policymakers need to take when determining where to initiate MDA against onchocerciasis. Our results suggest that there will be no hard threshold, but models—informed by sufficiently robust data—can assist with quantifying probabilities and timeframes of persistent transmission to help inform these decisions.

If feasible, longitudinal sampling of communities (i.e., multiple cross-sectional surveys of the same communities over time), using accurate and appropriate diagnostic tools would be crucial to both identifying communities with persistent transmission and ensuring sustained freedom of infection where elimination has been achieved. Typically, diagnostic specificity is prioritised over sensitivity for documenting elimination of infection. For example, the

specificity of some Ov16 ELISA protocols has been enhanced at the cost of sensitivity [11], to demonstrate, for instance, elimination at large scale in Latin America [53]. However, this approach either requires large sample sizes over assumed geographically homogenous transmission foci, or repeated sampling at community levels (in more heterogeneous settings) to compensate for reduced sensitivity [54]. In Africa, where progress towards elimination is patchy, repeated longitudinal sampling at community level, using highly sensitive and specific diagnostics, would be required both for the purposes of demonstrating sustained elimination and also for identifying persistent foci of low-level transmission.

In conclusion, we have explored, using EPIONCHO-IBM, the stability of onchocerciasis in low-transmission hypoendemic settings, characterising TBR ranges and the dynamics of persistence over epidemiologically-relevant timeframes. We emphasise that infection extinction in the absence of intervention must be seen as a probabilistic outcome, with no fixed epidemiological threshold below which transmission cannot be sustained [19]. Even very low mf prevalence can be sustained for long durations, and stability (or rate of spontaneous elimination) can only be quantified accurately with either concurrent data on ABRs or repeated epidemiological assessments. Consequently, low-transmission settings represent a serious impediment to EOT, as it cannot be assumed that they will fade out in acceptable timeframes without intervention, also presenting challenges when hypoendemic onchocerciasis foci are co-endemic with loiasis [12,45–47]. Therefore, in the absence of detailed epidemiological and entomological data to suggest otherwise, a conservative approach to reaching EOT is to initiate (appropriate) interventions wherever endogenous infection is confirmed, and to invest in and scale up the use of highly specific and sensitive diagnostic tools for surveillance to identify hypoendemic communities and ensure that EOT is sustained.

## Supporting information

**S1 Text.   Text A.** Brief description of EPIONCHO-IBM. **Table A.** Density dependence parameters determining parasite establishment within humans as a function of the annual transmission potential (ATP, no. L3/person/year) for different values of parameter $k_E$, of the gamma distribution describing inter-individual exposure heterogeneity. **Fig A.** Schematic representation of deterministic versus stochastic projections for low infection prevalence. **Fig B.** Initialising EPIONCHO-IBM with an endemic equilibrium. **Fig C.** Model-derived and fitted proportion of EPIONCHO-IBM runs persisting over time. **Text B.** Modelling for policy: PRIME-NTD. **Table B.** Policy-Relevant Items for Reporting Models in Epidemiology of Neglected Tropical Diseases (PRIME-NTD) summary table. **Fig D.** EPIONCHO-IBM predicted distribution of microfilarial (mf) prevalence over simulation time. **Fig E.** Influence of human population size ($N$) on model run persistence distributions. **Fig F.** Influence of inter-individual heterogeneity parameter, $k_E$, on persistence of transmission.
(DOCX)

## Acknowledgments

We thank Paul Cantey (US Centers for Disease Control and Prevention) and Katherine Gass (Task Force for Global Health) for invaluable discussions on the policy value of furthering our understanding of the stability of hypoendemic settings. We also thank James Truscott (Imperial College London) for helpful modelling and visualisation suggestions, and Raiha Browning (University of Warwick for insights into the modelled relationship between annual biting rates and exposure heterogeneity when fitting EPIONCHO-IBM to village-level mf infection data.

## Author contributions

**Conceptualization:** Jacob N. Stapley, Jonathan I. D. Hamley, Maria-Gloria Basáñez, Martin Walker.

**Data curation:** Jacob N. Stapley, Jonathan I. D. Hamley.

**Formal analysis:** Jacob N. Stapley, Jonathan I. D. Hamley, Martin Walker.

**Funding acquisition:** Jacob N. Stapley, Maria-Gloria Basáñez, Martin Walker.

**Investigation:** Jacob N. Stapley, Jonathan I. D. Hamley, Maria-Gloria Basáñez, Martin Walker.

**Methodology:** Jacob N. Stapley, Jonathan I. D. Hamley.

**Project administration:** Maria-Gloria Basáñez, Martin Walker.

**Resources:** Jacob N. Stapley, Jonathan I. D. Hamley, Maria-Gloria Basáñez, Martin Walker.

**Software:** Jonathan I. D. Hamley.

**Supervision:** Maria-Gloria Basáñez, Martin Walker.

**Validation:** Jonathan I. D. Hamley.

**Visualization:** Jacob N. Stapley, Jonathan I. D. Hamley, Maria-Gloria Basáñez, Martin Walker.

**Writing – original draft:** Jacob N. Stapley, Jonathan I. D. Hamley, Maria-Gloria Basáñez, Martin Walker.

**Writing – review & editing:** Jacob N. Stapley, Jonathan I. D. Hamley, Maria-Gloria Basáñez, Martin Walker.

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
