## [Decision Letter · Decision Letter 0]

5 Feb 2025

PCOMPBIOL-D-24-01739

Modelling transmission thresholds and hypoendemic stability for onchocerciasis elimination

PLOS Computational Biology

Dear Dr. Stapley,

Thank you for submitting your manuscript to PLOS Computational Biology. After careful consideration, we feel that it has merit but does not fully meet PLOS Computational Biology's publication criteria as it currently stands. Therefore, we invite you to submit a revised version of the manuscript that addresses the points raised during the review process.

Please submit your revised manuscript within 60 days Apr 07 2025 11:59PM. If you will need more time than this to complete your revisions, please reply to this message or contact the journal office at ploscompbiol@plos.org. Please include the following items when submitting your revised manuscript:

We look forward to receiving your revised manuscript.

Kind regards,

Emma Davis

Guest Editor

PLOS Computational Biology

Hannah Clapham

Section Editor

PLOS Computational Biology

**Journal Requirements:**

1) Please upload all main figures as separate Figure files in .tif or .eps format. For more information about how to convert and format your figure files please see our guidelines: 

**Reviewers' comments:**

Reviewer's Responses to Questions

Reviewer #1: I have had the chance to review the manuscript titled “Modelling transmission thresholds and hypoendemic stability for onchocerciasis elimination”. This manuscript addresses a crucial issue in the elimination of onchocerciasis, one of the neglected tropical diseases, particularly the dynamics in hypoendemic areas that have not been prioritized for mass drug administration (MDA). It employs the EPIONCHO-IBM stochastic individual-based model to explore the thresholds for transmission and the stability of infection under varying population and exposure heterogeneity conditions. The findings emphasize the need for extending treatment to all areas with endogenous infections. This manuscript can be published in PLOS Computational after the authors have addressed the following concerns.

• The authors have not compared the model’s outputs with real-world data or previous empirical findings, thereby limiting the ability to gauge its practical applicability. The authors need to incorporate comparisons with field data or historical studies to validate the findings and enhance their credibility.

• Can the authors enhance Hypoendemic stability findings by including case studies or field data? This can also be extended to include discussion of the implications of potential discrepancies between model predictions and observed data.

• The manuscript emphasizes stochastic dynamics but does not sufficiently address the variability across different ecological and social contexts, such as co-endemicity with other diseases. The authors need to address the implications of the model’s findings in co-endemic areas, especially where Loa loa is prevalent. This when addressed can provide a clear understanding into the endemicity of this disease.

• While the manuscript discusses the limitations of current diagnostic tools, it lacks concrete suggestions for integrating findings with Ov16 seroprevalence data or other metrics. The authors need to integrate results with potential field diagnostics, such as Ov16 seroprevalence or blackfly infection rates, to bridge the gap between modeling and surveillance.

• The manuscript does not sufficiently discuss the implications of assuming closed populations or the limitations in generalizing findings across diverse onchocerciasis-endemic regions. The authors also need to elaborate on the operational challenges and recommendations for implementing the study’s insights in low-resource settings.

• The authors need to simplify jargon in the discussion. As it is, the discussion is too technical for a broader audience, that could include policymakers and field practitioners.

Reviewer #2: This paper provides an in-depth exploration of the stochastic fade out in a well-established onchocerciasis transmission model. The simulations, statistical analysis and figures are very clear, and it is well written. The main result, that hypo-endemic transmission areas should not be missed by treatment campaigns, has important policy implications. My only comments are relatively minor:

1. I am not completely sold on the terminology. Borrowing the concept of thresholds from deterministic models and applying it to a stochastic model has it’s problems. This is rightly acknowledged in the limitations, there are no sharp thresholds. As such it is not clear why it is a useful concept, or at least, the manuscript doesn’t explicitly say why a threshold range is a useful concept. I think to strengthen the argument for defining it in this way, there should be an explicit description of how the system behaves (roughly) below, within, and above this range.

2. There should be some mention about the reason why the observed decay rate is slow, much slower than other helminthic diseases. I.e. that the assumed fertile lifespan of the adult female parasite is very long. This means that it may be more important to treat these hypo-endemic areas than for other helminths with shorter lifespans, where perhaps hypo-endemic local R0 < 1 transmission is more likely to be sustained by importations from higher transmission areas.

3. In the supplementary, the Fig A caption mentions a breakpoint, which normally refers to the unstable equilibrium for R0 > 1 created by positive density dependent mating. For consistency this should be changed to threshold.

Reviewer #3: Notes on Plos CompBio paper Stapley et al.

This is a well-written and interesting study using an existing stochastic model of onchocerciasis transmission to examine the stability of transmission in hypoendemic settings.

The results presented here challenge assumptions regarding transmission in hypoendemic settings, which raises important questions for onchocerciasis control policy guidelines if elimination of transmission is to be achieved. This points towards a need to dramatically review and expand the current targeting of elimination efforts.

Whilst there is clear value in studies that challenge existing policies and the assumptions that underly them, it is a little disappointing given the operational implications for this work that there is no clear evidence of engaging with stakeholders. It would really strengthen the authors’ position if they could demonstrate stakeholder engagement, either as co-authors or formally acknowledged through, for example, a PRIME-NTD table, and I would anticipate input from stakeholders to help ground both the scenarios modelled and the conclusions within the wider context of onchocerciasis control efforts.

Other points:

1. Why are 500 year simulations used? Why 300 model runs? 500 years doesn’t seem to be an epidemiologically relevant timescale, and it would be more interesting to see a shorter timescale examined in more detail.

2. What informed the values of kE used (0.2-0.4)? Were these derived from fitting to data?

3. Given the impact of population size on stochastic processes/fade out, why were population sizes of N=50-1000 used? Was this informed by demographic data from endemic areas?

4. How could kE and/or ABR be expected to change under interventions? It would be interesting to see simulations with more varying levels of heterogeneity, and given the challenges of MDA in some hypoendemic settings due to co-endemic loiasis, it would be interesting to see simulations of interventions targeting the vector, which may reduce ABR/impact heterogeneity.

**Have the authors made all data and (if applicable) computational code underlying the findings in their manuscript fully available?**

Reviewer #1: Yes

Reviewer #2: Yes

Reviewer #3: Yes

PLOS authors have the option to publish the peer review history of their article (what does this mean? ). If published, this will include your full peer review and any attached files.

**Do you want your identity to be public for this peer review?** For information about this choice, including consent withdrawal, please see our Privacy Policy .

Reviewer #1: No

Reviewer #2: No

Reviewer #3: No

**Figure resubmission:**
---

## [Editor Report · Decision Letter 1]

7 Apr 2025

Dear Dr Stapley,

We are pleased to inform you that your manuscript 'Modelling transmission thresholds and hypoendemic stability for onchocerciasis elimination' has been provisionally accepted for publication in PLOS Computational Biology.

Best regards,

Emma Davis

Guest Editor

PLOS Computational Biology

Hannah Clapham

Section Editor

PLOS Computational Biology

---

## [Editor Report · Acceptance letter]

PCOMPBIOL-D-24-01739R1

Modelling transmission thresholds and hypoendemic stability for onchocerciasis elimination

Dear Dr Stapley,

I am pleased to inform you that your manuscript has been formally accepted for publication in PLOS Computational Biology. Your manuscript is now with our production department and you will be notified of the publication date in due course.

With kind regards,

Anita Estes
